**Plant Genetics and Genomics**

# Targeted seed EMS mutagenesis reveals a basic helix–loop–helix transcription factor underlying male sterility in sorghum

Yuguo Xiao,[1,†] Rajdeep S. Khangura [ID],[2,3,†] Zhonghui Wang,[1] Brian P. Dilkes [ID],[2,3] Andrea L. Eveland [ID] [1,*]

[1]Donald Danforth Plant Science Center, St. Louis, MO 63132, USA
[2]Department of Biochemistry, Purdue University, West Lafayette, IN 47907, USA
[3]Center for Plant Biology, Purdue University, West Lafayette, IN 47907, USA

*Corresponding author: Donald Danforth Plant Science Center, St. Louis, MO 63132, USA. Email: AEveland@danforthcenter.org
†Y.X. and R.S.K. contributed equally to this work.

Forward genetic screens of mutant populations are fundamental for functional genomics studies. However, isolating independent mutant alleles to molecularly identify causal genes is challenging in species recalcitrant to genetic manipulation. Here, we demonstrate that classic seed ethyl methanesulfonate (EMS) mutagenesis coupled with genome sequencing can overcome this limitation in sorghum. We used this method to generate new mutant alleles of sorghum *MALE STERILE 8* (*MS8*) and identified the causal locus for the *ms8* phenotype as *Sobic.004G270900*, which encodes the sorghum ortholog of maize *bhlh122*, a basic helix–loop–helix (bHLH) transcription factor required for male fertility in maize. Bulked segregant analysis mapped *ms8-1* to a region on chromosome 4 containing *Sobic.004G270900*. Seeds from heterozygous *MS8*/*ms8-1* plants were mutagenized and screened for chimeric inflorescences containing sectors with white, sterile anthers resembling the *ms8-1* homozygous phenotype. DNA sequencing of sterile and fertile sectors from a single chimeric inflorescence revealed two mutations in *Sobic.004G270900* within the sterile sector, but not the fertile sector. Isolation of this loss-of-function allele (*ms8-2*) established *Sobic.004G270900* as the causative locus for male sterility in the *ms8* mutant. We generated additional alleles of *MS8* in a different genetic background using CRISPR/Cas9-based gene editing, where deletions in *Sobic.004G270900* also resulted in male sterility. Our work identified a gene underlying male sterility in sorghum and provides a novel and straightforward genetic tool for researchers who lack access to advanced transformation facilities to validate gene candidates. Unlike gene editing, no prior knowledge of candidate genes is required for targeted seed EMS mutagenesis to aid identification of causal loci.

Keywords: bulked segregant analysis; male sterility; chemical mutagenesis; sorghum; germinal sectors; noncomplementation

## Introduction

Mutant populations are valuable resources for functional genomics, enabling causative mutations to be linked to phenotypes of interest through mapping and/or sequencing (Jiang and Ramachandran 2010; Jiao *et al.* 2017; Wang *et al.* 2021; Gupta *et al.* 2023; Xiong *et al.* 2023). Forward genetic screens using mutant populations often yield a single mutant allele for a phenotype of interest. Reverse genetics resources can be used to validate gene candidates identified through forward genetics or other methods but the number of lines sequenced is often limited, and multiple loss-of-function alleles may not be recovered for a given gene. Due to the high rate of background mutations in experimental mutagenized populations, it is necessary to identify independent mutant alleles to distinguish a true causative mutation in a gene from linked noncausative mutations. Reverse genetics tools such as TILLING (McCallum *et al.* 2000), or gene editing tools such as Cas9-guide (g)RNA approaches (van der Oost and Patinios 2023) can be used to isolate independent alleles in a candidate gene but they are resource-intensive, not available in all species, take at least two generations for validation, and most importantly, require knowledge of candidate genes beforehand.

In this study, we present a forward genetic mutagenesis screen in sorghum (*Sorghum bicolor*) specifically designed to identify novel alleles of mutants with visible phenotypes. We couple a classic strategy termed "targeted mutagenesis" with linkage mapping and genome sequencing. Targeted mutagenesis gets its name not because of specific targeting of DNA but because of the experimental crossing design that facilitates recovery of loss-of-function mutants at a particular locus. For recessive mutants, mutagenesis is carried out on heterozygous material in an effort to recover new alleles by screening for a failure to complement when the allele is mutated. This approach is popular in maize by mutagenizing wild-type pollen and using that to pollinate mutant testers, followed by screening of the F₁ generation to recover novel mutant alleles that fail to complement (Candela and Hake 2008). We have repeatedly successfully taken this approach to recover novel alleles when only one was recovered in our primary genetic screen (Marla *et al.* 2018; Best *et al.* 2021; Sauer *et al.* 2023). The pollen mutagenesis approach is highly efficient in monoecious plants like maize where male and female inflorescences are distinct. This approach, though highly effective, suffers from few drawbacks such as recovery of false positives due to occasional self-pollination of the mutant tester or

gynogenic haploids (Best *et al.* 2021). A similar targeted screen can be done by mutagenizing heterozygous seeds to recover new loss-of-function alleles visible as somatic sectors (Candela and Hake 2008). This technique can generate somatic leaf chimeras resulting from loss of function in dominant alleles. For example, mutagenesis of heterozygotes at the semidominant lesion mutant *Rp1-D21* in maize generated derivative suppressor alleles of *Rp1-D21* that were visible as wild-type leaf sectors on an otherwise lesioned *Rp1-D21/+* heterozygous plant (Karre *et al.* 2021).

Here, we deployed a targeted genetics mutagenesis screen to isolate novel mutant alleles and accelerate molecular identification of genes responsible for mutant phenotypes in sorghum. Male sterility is an agronomically important trait for hybrid seed production. In sorghum, several recessive nuclear male sterility mutant loci (*msal*, *ms1*, *ms2*, *ms3*, *ms7*, *ms8*, and *ms9*) are known (Pedersen and Toy 2001; Xin *et al.* 2017; Chen *et al.* 2019). Only *male sterile9* has been molecularly identified and encodes a PHD-finger transcription factor (Chen *et al.* 2019). A mutant allele at the *MALE STERILE8* (*MS8*) locus, hereafter referred to as *ms8-1*, was morphologically characterized previously (Xin *et al.* 2017, 2018), but its map position and gene identity were unknown. This allele exhibits a recessive inheritance, and the homozygous *ms8-1* plants produce inflorescences with florets containing bright white anthers at anthesis that are completely devoid of pollen due to disruption of tapetum development (Pedersen and Toy 2001; Xin *et al.* 2017; Chen *et al.* 2019). We hypothesized that loss of the wild-type allele in a sector of a *MS8/ms8-1* heterozygous head would result in white anthers in an otherwise fertile inflorescence at anthesis. Using this approach, we isolated a novel mutant allele, *ms8-2*, that provided evidence of the underlying gene. We further validated the gene by generating additional alleles using a CRISPR/Cas9-based gene editing approach in a different genetic background. Together this demonstrates that *Sobic.004G270900* encodes *MS8*. This gene is orthologous to maize *bhlh122* (Nan *et al.* 2022), which encodes a basic helix–loop–helix (bHLH) transcription factor required for male fertility and tapetum development in multiple plant species (Niu *et al.* 2013; Ji *et al.* 2013; Zhu *et al.* 2015; Nan *et al.* 2022).

The targeted seed ethyl methanesulfonate (EMS) mutagenesis approach described here is simple, time-saving and relies on built-in complementation testing within the same generation as part of the experimental design. This offers a significant advantage in species such as sorghum where multiple mutagenized populations (Addo-Quaye *et al.* 2018; Jiao *et al.* 2024) enable functional genomics studies to identify gene-to-phenotype relationships, but oftentimes lack independent mutant alleles required to validate candidate genes. Furthermore, this approach can be easily adapted to characterize mutants displaying any visible cell autonomous phenotype at the whole plant level in other plant species. This makes it particularly valuable for species or cultivars that are either recalcitrant to transformation protocols for gene editing, have long generation times, or lack resources altogether.

## Materials and methods
### Plant materials and genetic nomenclature
The *ms8-1* mutant allele was isolated from a mutagenized BTx623 population described previously (Xin *et al.* 2017; Zhao and Xin 2019). We obtained the seeds of *ms8-1* genetic stock in BTx623 background from the laboratory of Dr. Gebisa Ejeta at Purdue. Seeds of ms8-1 genetic stock in BTx623 are freely available from GRIN (Xin *et al.* 2018). All gene, allele, and gene product names are provided here in the format detailed by Meinke and Koornneef (1997) in their guidelines for the Arabidopsis genetics

community as this is the most common format used in plant genetics, and there is no accepted community standard for sorghum.

### Bulked segregant analysis mapping of *ms8*
We performed bulked segregant analysis (BSA) analysis to map the *ms8-1* mutation in sorghum, following the method described in Klein *et al.* (2018) with some adjustments. Two independent *ms8-1* $F_2$ mapping populations were generated by crossing pollen from other EMS mutagenized BTx623 lines (Addo-Quay *et al.* 2018) onto *ms8-1* heads, and selfing the $F_1$ *MS8/ms8-1* plants. These $F_2$ populations were planted at the Danforth Center's Research Field in the summer of 2022 and phenotyped for male sterility. DNA was extracted using a CTAB method (Gallavotti and Whipple 2015) from pooled leaf tissue collected from male sterile and male fertile plants, respectively. Each pooled sample was made by taking four leaf punches from each individual using a paper punch. Sequencing libraries were prepared using the NEBNext Ultra II FS DNA Library Prep Kit for Illumina (E7805) according to the manufacturer's instructions (NEB). Sequencing of the DNA pools was performed on an Illumina Hi- Seq 2500 at Novogene Corporation, generating 150-bp paired-end reads. FastQC (v0.11.9) was used to assess the overall quality of the sequencing data ("Babraham Bioinformatics—FastQC A Quality Control tool for High Throughput Sequence Data"). Trimmomatic (v0.35) (Bolger *et al.* 2014) was used to filter and remove low-quality reads and adapter sequences with the following parameters: ILLUMINACLIP: TruSeq3:2:30:10 LEADING:25 TRAILING:25 SLIDINGWINDOW:5:20 MINLEN:36. The trimmed reads were mapped to version 3 of the sorghum reference genome BTx623 (Sbicolor_454_v3.0.1; McCormick *et al.* 2018), using Bowtie2 (v2-2.2.9; Langmead and Salzberg 2012) with the following parameters: –maxins 2000 –dovetail. Only uniquely mapped reads with a mapping quality greater than 20, determined by SAMtools (v1.11), were used for subsequent variant calling.

SAMtools and BCFtools were used to generate pileup and variant call format (VCF) files from the indexed BAM file (Li *et al.* 2009). Pileup files were used to identify the mapping region, while VCF files were used to identify candidate SNPs. The pileup and VCF files were first filtered out to remove nonvariant positions to focus on variant positions with a minimum coverage of two reads. Next, the pileup and VCF files from the *ms8-1* DNA pool were further filtered to exclude homozygous variants shared with its wildtype sibling DNA pool and experimentally determined error-prone sites shared among individuals in the same genetic background (Addo-Quaye *et al.* 2018). The final filtered pileup file was split into 10 chromosomes and plotted against the BTx623 reference genome to identify chromosomal regions enriched for homozygous SNPs in the mutant pool due to linkage to the causative mutations. In this analysis, homozygous variants were defined as nucleotide positions that differed from the reference genome at a frequency greater than or equal to 0.99. The boundaries of the *ms8-1* mapping intervals were defined as the chromosomal coordinates where variant frequency returned to background levels.

To identify potential causative lesions within the mapping intervals, SnpEff (v4.3a) was used to predict the impact of SNPs on gene function (Cingolani *et al.* 2012). We filtered out SNPs that were not homozygous or noncanonical EMS changes (G to A or C to T). Candidate SNPs were subsequently validated through Sanger sequencing.

### Phylogenetic analysis of bHLH transcription factors
Phylogenetic tree construction of bHLH transcription factors followed the protocol outlined by Bélanger *et al.* (2023), with some

modifications. Proteome sequence files were obtained from Phytozome (https://phytozome-next.jgi.doe.gov/), except for *Arabidopsis thaliana*, which was sourced from TAIR (https://www.arabidopsis.org/). To label individual proteins in the phylogenetic tree, species names were abbreviated and combined with protein IDs.

OrthoFinder v2.5.4 (Emms and Kelly 2015) was used to identify orthologous protein groups among eight species using default parameters. Orthologous groups for the bHLH family were identified using the known bHLH protein sequences from Arabidopsis (*A. thaliana*), rice (*Oryza sativa*), and maize (*Zea mays*) obtained from the Plant Transcription Factor Database (PlantTFDB v5.0, https://planttfdb.gao-lab.org/download.php). The presence of the bHLH domain in protein sequences was confirmed by searching Pfam r35.0 (Mistry *et al.* 2021) using hmmscan from HMMER v3.3.2 (http://hmmer.org) with parameter -E 0.00001, followed by manual curation using CDvist (Adebali *et al.* 2015). Unaligned homologous sequences were inspected, and nonhomologous adjacent characters were removed using PREQUAL v1.02 (Whelan *et al.* 2018) with default parameters. Multiple sequence alignments were generated using MAFFT v7.505 (Katoh and Standley 2013) with parameters –auto –anysymbol –dash –originalseqonly.

Protein alignments were trimmed using trimAL v1.4.1 (Capella-Gutiérrez *et al.* 2009) with parameters -gt 0.9 -cons 60 -w 3 to remove nonhomologous regions outside the bHLH domains. Maximum-likelihood phylogenetic tree was inferred using IQ-TREE v2.2.0.3 (Nguyen *et al.* 2015) with parameters -B 1000 -mset JTT -T 32 –seqtype AA. The resulting bHLH phylogeny was visualized on iTOL v6 (https://itol.embl.de/), with clades distant from the sorghum MS8 clade collapsed for clarity.

## Generation of additional *ms8* alleles by targeted EMS mutagenesis

Sorghum seed for chemical mutagenesis was created by crossing the male sterile *ms8-1/ms8-1*:BTx623 plants with pollen from *MS8/ms8-1*:BTx623 plants. The $F_1$ seed from this cross was treated with 30 mM EMS (Sigma-Aldrich catalog# M0880) dissolved in water. Briefly, on the day before planting, dry sorghum seeds from this $F_1$ cross were soaked in 30 mM EMS solution for 12 hours in the dark. The EMS solution was decanted into a waste container and seeds were rinsed six times with tap water. After the last rinse, these mutagenized ($M_1$ generation) seeds were laid out on absorbent paper in a fume hood for three hours at room temperature to dry.

The dried $M_1$ seeds were treated with a slurry of NipsIt INSIDE insecticide (Valent U.S.A LLC, San Ramon, CA, USA), Apron XL (fungicide, Syngenta, Basel, Switzerland), Maxim XL (fungicide, Syngenta, Basel, Switzerland), and a grass-specific herbicide seed safener Concep III (Syngenta, Basel, Switzerland) and then dried for an additional 4 h. After fungicide and safener treatment, the $M_1$ seeds were packed into plot packets with ~37 seeds per packet. The $M_1$ generation was planted at Purdue University's Agronomy Center for Research and Education farm in the summer of 2023. Each packet was planted in the field as a 15-foot plot, with 12.5 feet of planted space and 2.5 feet of alley between successive plots. The spacing between adjacent rows was fixed at 2.5 feet. This spacing achieved a planting density of ~45,000 plants per acre. A final plant count of ~2,500 $M_1$ plants was established and subsequently screened at anthesis to identify additional alleles of *ms8* as chimeric plants with white anthers on an otherwise fertile inflorescence with yellow anthers.

## Bioinformatics analysis of the male sterile and male fertile sectors

DNA was isolated from male sterile and male fertile florets collected from a chimeric inflorescence using a modified CTAB method (Saghai-Maroof *et al.* 1984). Whole-genome shotgun sequencing was performed using the NovaSeq X Plus platform (Illumina Inc., San Diego, CA, USA) by the Novogene Corporation (Sacramento, CA, USA). Paired-end reads ($2 \times 150$ bp) were processed to remove low-quality base calls and adapter sequences using *fastp* (Chen *et al.* 2018). The cleaned paired-end reads were aligned to the BTx623 v3.0.1 genome (McCormick *et al.* 2018) with the *bwa-mem* aligner using the default parameters (Li and Durbin 2009). Sorted BAM files were used to call SNPs using *mpile-up* functions in *bcftools* (Danecek *et al.* 2021). As EMS primarily causes $G > A$ and $C > T$ transitions, the SNP calls were filtered to retain these changes using an *awk* command. Subsequently, variant effect prediction using *SnpEff* (Cingolani *et al.* 2012) identified moderate to high impact changes in the protein-coding regions. The sorted BAM files were visualized in IGV (Robinson *et al.* 2011).

## Generation of additional *ms8* alleles by gene editing

To generate additional *ms8* mutant alleles, gene-specific guide (g)RNA sequences were designed using CRISPR-P 2.0 (http://crispr.hzau.edu.cn/cgi-bin/CRISPR2/CRISPR) and cloned into pMOD_2303 (Čermák *et al.* 2017). Subsequently, the gRNA expression cassettes were transferred into a plant binary CRISPR–Cas9 transformation vector coexpressing Cas9 and morphogenic regulators (Lowe *et al.* 2016). The final vector was transformed into the sorghum Tx430 genetic background using an *Agrobacterium*-mediated protocol (Wang *et al.* 2023) in the Eveland lab at the Donald Danforth Plant Science Center. Hygromycin selection followed by PCR of the transgene was used to identify positive transgenic $T_0$ events. In total, 15 independent $T_0$ events were identified. $T_1$ plants were screened for absence of the transgene and then Cas9-free lines screened for stable edits by DNA sequencing. Primers used for vector constructions are listed in Supplementary Table 1.

## Results

### BSA maps *ms8-1* to chromosome 4

*ms8-1* is a recessive male sterile mutant in sorghum with small, white, and nondehisced anthers (Fig. 1a and b). We used BSA to map the causal lesions responsible for the *ms8-1* phenotype (Fig. 1c). Two pooled DNA samples, *ms8-1* pool1 and *ms8-1* pool2 (Supplementary Table 2), containing 57 and 37 *ms8-1* homozygous individuals from two independent $F_2$ mapping populations, respectively, were sequenced on an Illumina platform. To eliminate background variants from the populations, we sequenced two pooled DNA samples, containing 40 and 42 wild-type sibling individuals from the respective $F_2$ populations. Sequencing yielded a total of around 350 million paired-end raw reads, with 62.4 to 101.9 million reads per library. Approximately 96% of reads mapped to the BTx623 v3 reference genome. Mapped reads were filtered to remove low-quality alignments and PCR duplicates prior to variant calling. This resulted in genomic coverages from 8.9- to 14.3-fold per pooled DNA sample (Supplementary Table 2).

To identify the chromosomal regions linked to the *ms8-1* phenotype, we searched for genomic regions enriched for homozygous variants in the two *ms8-1* mutant pools. We found 5,711 and 4,400 homozygous variants (allele frequency $\geq$ 0.99) in the *ms8*-pool1 and *ms8*-pool2 datasets, respectively. We then filtered out putative background variants and error-prone positions in these mutant pools following procedures outlined in the Methods. After filtering, 48 and 101 homozygous SNPs remained in the *ms8-1* pool1 and *ms8-1* pool2 data, respectively (Supplementary Table 3). Plotting the final filtered homozygous variants across the sorghum genome revealed a cluster of homozygous variants on chromosome 4

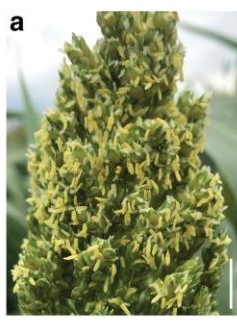
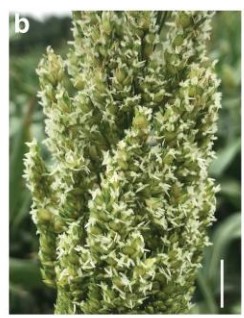

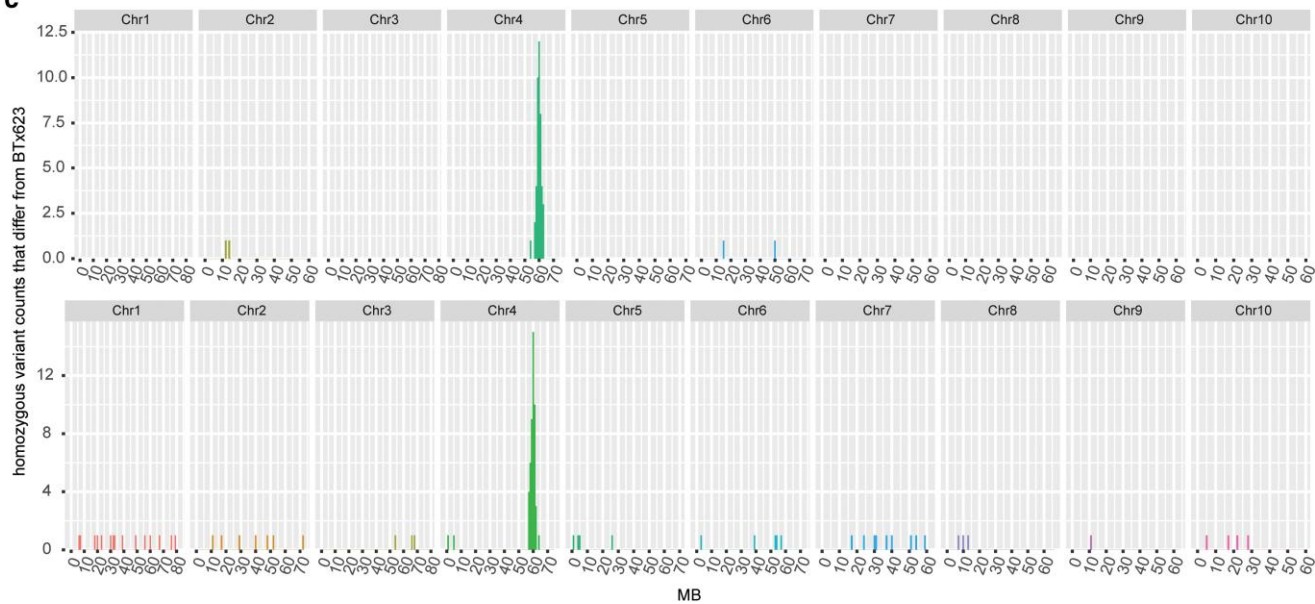

**Fig. 1.** BSA mapping localized *ms8-1* to a region on chromosome 4. a) Wild-type panicle with yellow and fertile florets. b) *ms8-1* panicle with white, male sterile florets. c) BSA identified the chromosomal location of the causative mutation(s) in the *ms8-1* mutant. The number of nonreference homozygous variants (variant frequency ≥ 0.99) per 1 megabase pair (Mbp) chromosomal bin was plotted across all chromosomes for two independent *ms8-1* mutant pools: *ms8*-pool1 (upper panel) and *ms8*-pool2 (lower panel). A prominent peak is observed in both pools within the same genomic region (between 56 and 64 Mbp on chromosome 4), suggesting that *ms8-1* lies within this region. Scale bars represent 2 cm.

**Table 1.** Protein-coding variants within the mapped interval for ms8-1.

| Chr | Position | Ref | Alt | Variant type | Impact | Effect | Gene ID | Functional annotation |
|---|---|---|---|---|---|---|---|---|
| Chr4 | 59914160 | C | T | missense_variant | MODERATE | p.D209N | Sobic.004G253400 | TIC (translocation at the inner envelope membrane of chloroplasts) homologous protein |
| Chr4 | 60155882 | C | T | missense_variant | MODERATE | p.G433D | Sobic.004G255500 | Rubisco methyltransferase family protein |
| Chr4 | 61494009 | C | T | stop_gained | HIGH | p.Q150* | Sobic.004G270900 | Basic helix–loop–helix (bHLH) DNA-binding superfamily protein |
| Chr4 | 61731674 | C | T | missense_variant | MODERATE | p.L478F | Sobic.004G273500 | Oligopeptide transporter 4 |
| Chr4 | 61914899 | C | T | missense_variant | MODERATE | p.G152E | Sobic.004G275800 | Plant U-box 23 |

between 56 and 64 Mbp in both mutant pools (Fig. 1c). This indicated that *ms8-1* tightly cosegregated with this genomic interval. This initial *ms8-1* mapping interval encodes 968 genes.

We next looked to identify potential causative variants responsible for the *ms8-1* phenotype. Given the considerable depth of sequencing for the two *ms8-1* mutant pools (Supplementary Table 2), our datasets were likely to include causative lesions. Among the 48 and 101 homozygous SNPs detected in *ms8*-pool1 and *ms8*-pool2, 28 were shared in both pools (Supplementary Table 3). These 28 shared SNPs were located within an overlapping region that narrowed the candidate interval to 58.3–62 Mbp (Supplementary Table 4). Using SnpEff (Cingolani *et al.* 2012), we evaluated the potential impact of these 28 variants on gene function. This identified five variants, four missense variants and one stop-gain

variant that alter the protein-coding sequence of a gene in the mapped window (Table 1). The stop-gain variant, located at genomic position 61,494,009, introduced a premature stop codon (448C > T, Q150*) within a gene encoding a bHLH transcription factor (*Sobic.004G270900*). Phylogenetic analysis revealed that *Sobic.004G270900* encodes the sorghum ortholog of maize bHLH122, rice EAT1/DTD1, and Arabidopsis bHLH010/089/091 (Fig. 2). All three of these transcription factors are established regulators of anther development and knockout mutants exhibited nondehisced anthers and tapetum development defects (Niu *et al.* 2013; Ji *et al.* 2013; Zhu *et al.* 2015; Nan *et al.* 2022) strikingly similar to the *ms8-1* phenotype (Xin *et al.* 2017). We hypothesized that *ms8-1* is a loss-of-function allele of *Sobic.004G270900* resulting in male sterility.

**Fig. 2.** Phylogeny of bHLH proteins in several cereal crops and Arabidopsis. The phylogeny is focused on male sterility-associated bHLH clades for clarity. Clades distantly related to male sterility-associated bHLH clades were collapsed. Species included in the analysis are maize (*Zea mays*; B73), sorghum (*Sorghum bicolor*; BTx623), rice (*Oryza sativa japonica*), barley (*Hordeum vulgare Morex*), wheat (*Triticum aestivum*), foxtail millet (*Setaria italica*), and *Arabidopsis thaliana*. Protein names are prefixed with Zm (maize), Sobic (sorghum), LOC (rice), HORVU.MOREX.r3 (*H. vulgare Morex*), Treas (wheat), Seita (*S. italica*), and AT (*A. thaliana*), respectively.

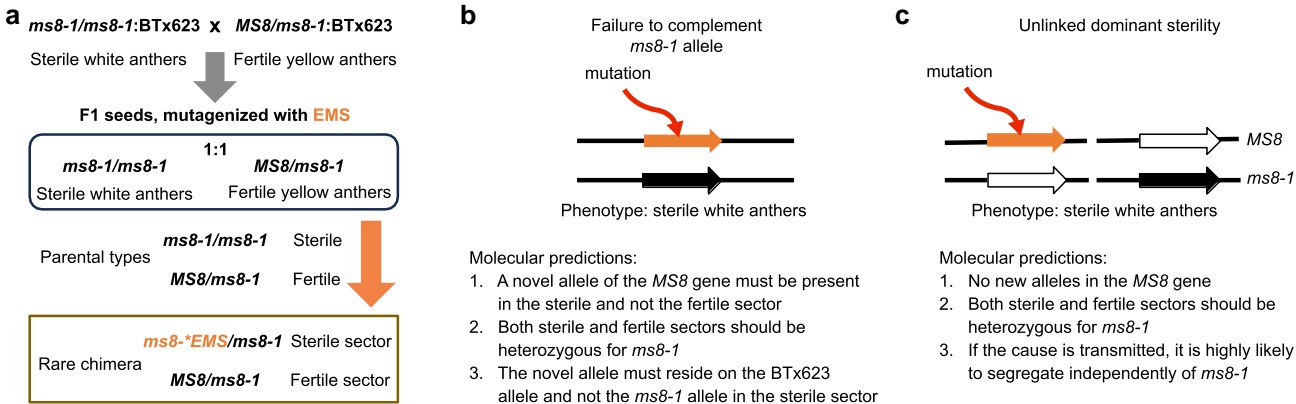

**Fig. 3.** Experimental design and molecular predictions of targeted seed mutagenesis. a) Experimental design of targeted seed mutagenesis to induce chimeras resulting from novel loss-of-function alleles of the *MS8 gene*. Molecular predictions for sterile sectors on a mutagenized male fertile *MS8/ms8-1* inflorescence are shown, resulting from b) mutation in the wildtype BTx623 *MS8* allele and c) a novel dominant mutation or chromosomal break.

## Isolation of independent alleles of *ms8* using a targeted EMS mutagenesis approach

The white anther phenotype of *ms8-1* homozygotes is easy to distinguish from the yellow, plump anthers on fertile florets. We reasoned that seed mutagenesis of *MS8/ms8-1* heterozygotes should produce chimeric heads with both fertile and sterile sectors when EMS knocks out the wildtype allele. The fertile heads are the expected phenotype of *MS8/ms8-1* heterozygote while sterile sectors may result from novel loss-of-function mutations within the wildtype *MS8* allele that fail to complement the *ms8-1* allele (Fig. 3). We pollinated male sterile *ms8-1* homozygotes with pollen from *MS8/ms8-1* heterozygotes. This cross produced progeny with

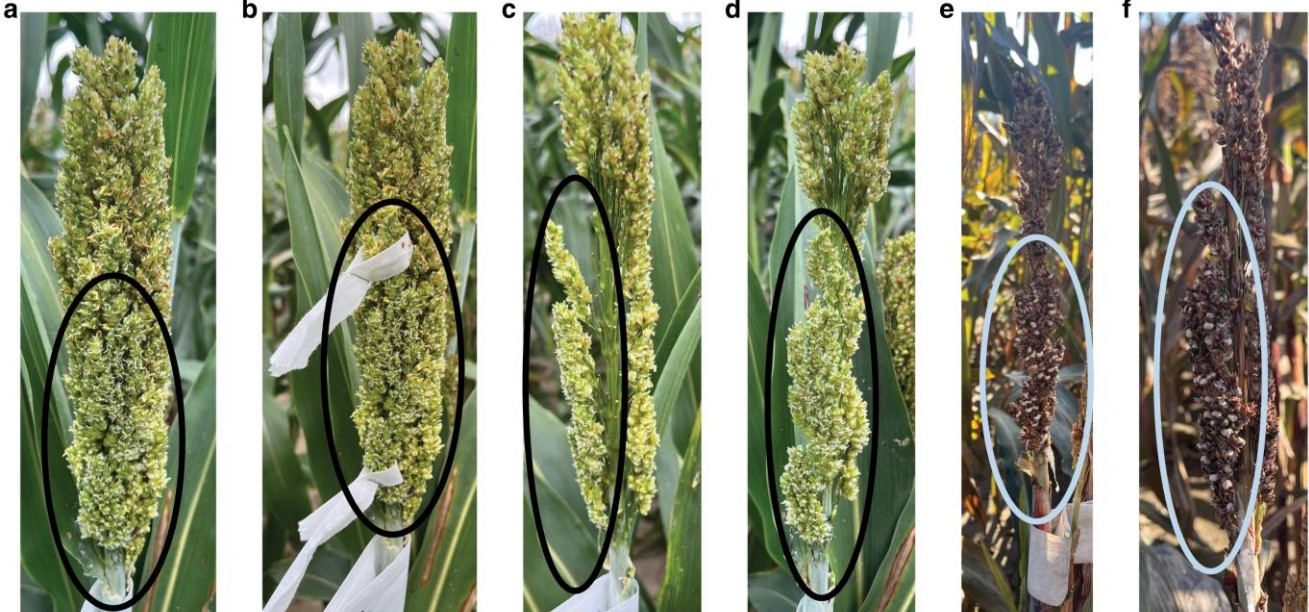

**Fig. 4.** A chimeric inflorescence with a male sterile sector identified in an M1 *MS8/ms8*-1 plant mutagenized by seed treatment with EMS. The male sterile florets (within the dashed ovals) are evident at anthesis (a–d) on an otherwise male fertile inflorescence. Sterility was overcome by supplemental pollination as evident at maturity (e and f).

an expected 1:1 segregation ratio of male-fertile *MS8/ms8*-1 and male-sterile *ms8*-1/*ms8*-1 individuals. Approximately 4,000 seeds from this cross were mutagenized with 30 mM EMS and planted (M$_1$ generation) (Fig. 3a). The resulting M$_1$ population of ~2,500 plants displayed varying degrees of spontaneous, likely dominant, somatic mutated sectors on their leaves, confirming the efficacy of this seed EMS mutagenesis. Only the *MS8/ms8*-1 heterozygotes in this M$_1$ population were useful for our purpose as they are fully fertile and carry one functional wild-type *MS8* allele and one loss-of-function *ms8*-1 allele. We reasoned that if part of the progenitor cells giving rise to the sorghum florets carried an EMS-induced disruptive mutation in the *MS8* wild-type allele, complementation to the *ms8*-1 allele will not occur, resulting in a sector of florets with sterile anthers on an otherwise fertile inflorescence (Fig. 3b). This sterility should be readily overcome by supplemental pollination with wild-type pollen. In contrast, sterility caused by chromosomal aberrations or gametophytic lethality is less likely to be overcome by supplemental pollination (Fig. 3c). Regardless of the cause (lack of complementation at *ms8*-1 or other factors), sterile sectors resembling the *ms8*-1 homozygote phenotype (white and nondehiscent anthers) should be visible on an otherwise male fertile head (yellow and plump anthers) during anthesis.

During anthesis, the M$_1$ population was scouted daily between 9 AM and 11 AM for chimeric heads containing a sector of the inflorescence producing florets with white anthers on a head otherwise bearing florets with fertile yellow anthers. Sterile florets were provided with supplemental pollen to test if they phenocopied the *ms8*-1 mutants. If the sterile florets produced seed, the sector was considered a candidate for a new allele of *ms8*. We identified a single fertile sorghum head with a sector displaying white anthers, resembling *ms8*-1/*ms8*-1 homozygotes, on an otherwise fertile head (Fig. 4a). The white anthers in this sector were shriveled and did not show any sign of pollen production. The remainder of the inflorescence in this sorghum head displayed fertile pollen with plump, yellow anthers. To isolate the sterile sector, the

florets with the white anther phenotype were tagged (Fig. 4b) and the surrounding fertile florets were removed (Fig. 4c and d). Supplemental pollination was then carried out using wild-type BTx623 pollen to assess whether the sterility was limited to the anthers. At maturity, this male sterile sector produced 40 seeds (Fig. 4e and f), demonstrating these florets were female-fertile.

To identify the causative mutation underlying this male sterile sector, DNA was extracted from tissue sampled from the sterile and fertile sectors of the sorghum head and sequenced separately at approximately 60× coverage (Supplementary Table 5). Variants not present in the reference genome were identified for each DNA pool following the procedures outlined in the Methods. The purpose of sequencing is to identify a potential novel allele of *MS8* gene which would be present in a heterozygous state affecting the wild-type *MS8* allele in the sterile sector but not in fertile sectors (Fig. 3). Since the *MS8* gene was already mapped within 58–63 Mb on chromosome 4, all heterozygous variants within this genomic region in both sterile and fertile sectors were identified. The SNP calls from the whole-genome sequencing data from the sterile sector contained 154 G-to-A or C-to-T changes also present in the fertile sector. All EMS-induced homozygous mutations identified in the *ms8*-1 homozygous pools BSA were present in both the fertile and sterile sectors of this sorghum head at an allele frequency of ~50% (Supplementary Table 6). This demonstrated that this individual was an *MS8/ms8*-1 heterozygote before EMS mutagenesis. The SNP calls from the whole-genome sequencing data from the sterile sector contained 87 G-to-A or C-to-T changes that were not present in the fertile sector. Only one gene with coding sequence differences in *ms8*-1 (Table 1), *Sobic.004G270900*, carried novel mutations that were unique to the male sterile sector (Fig. 5). This gene contained two novel C > T transitions in the sterile sector. The first C > T transition at 61493892 bp resulted in a P98S missense mutation, and a C > T transition at 61493853 bp resulted in a Q111* premature stop codon. The two mutations were linked in cis and all reads that overlapped the two sites either had both P98S and Q111* mutant

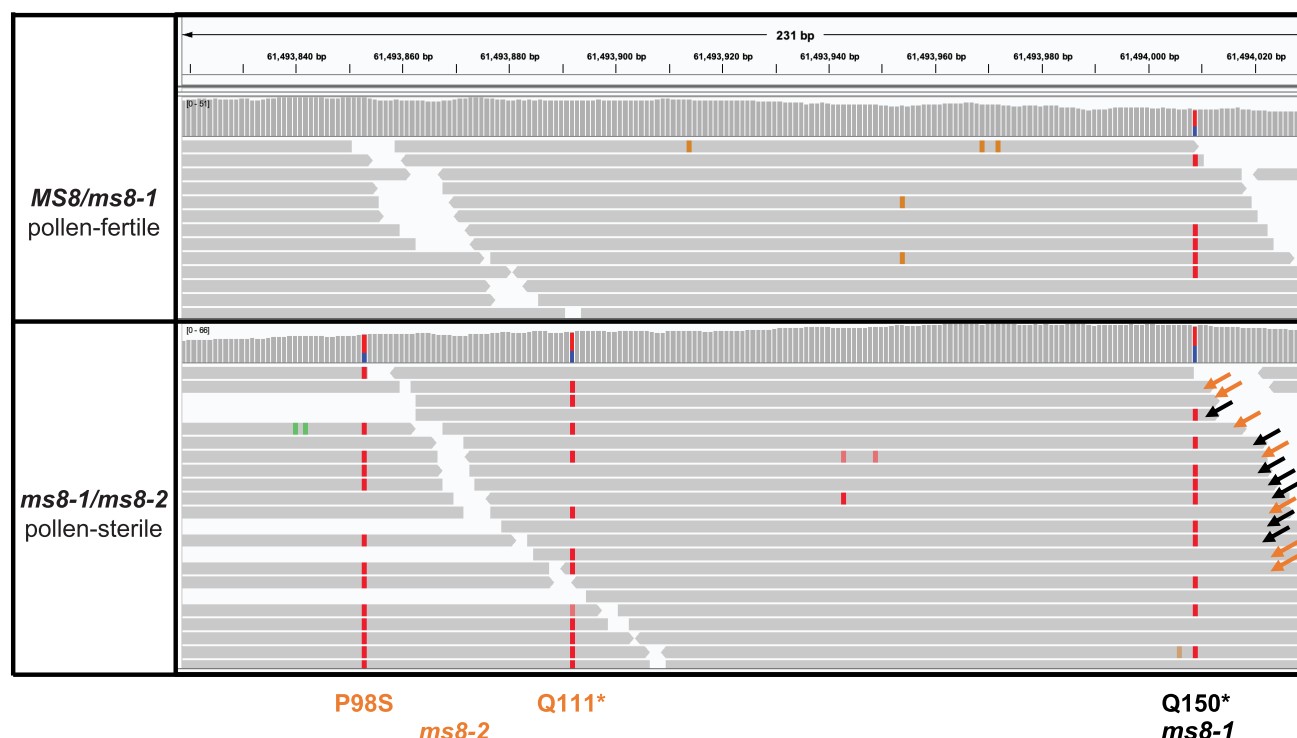

**Fig. 5.** Identification of a novel loss-of-function allele of the *MS8* gene by sector analysis. The IGV browser view shows the alignments of reads from the male fertile (*MS8*/*ms8-1*) and male sterile (*ms8-1*/*ms8-2*) sectors of the same inflorescence to the reference genome. Orange and black arrows represent reads mapped to the wildtype *MS8* allele and the *ms8-1* allele, respectively. Red rectangles indicate mutations that differ from the wildtype *MS8* allele.

alleles in the male-sterile sector or both wild-type alleles at these positions (Fig. 5). The C > T change identified by BSA that resulted in a Q150* nonsense mutation in *Sobic.004G270900* was present as a heterozygote in both the male sterile and male fertile sectors. The presence of a male-sterile sector that was fertile with supplemental pollination demonstrates the reproductive defect was male specific due to the failure of *ms8-2* (P98S and Q111*) to complement *ms8-1* (Q150*) and demonstrates that *MS8* is encoded by Sobic.004G270900.

MS8/*ms8-1* heterozygotes contain two distinct alleles of the *Sobic.004G270900* gene: a wild-type allele and a recessive loss-of-function *ms8-1* allele (Q150*). Careful examination of the sequence data in the male sterile sector confirmed that *ms8-1* and *ms8-2* were encoded on the two homologous chromosomal copies of *Sobic.004G270900*. The P98S and Q111* mutations were never detected on the copy of *Sobic.004G270900* encoding the *ms8-1* allele (Fig. 5 and Supplementary Fig. 1). All reads overlapping Q150 and Q111 were either Q150* or Q111* (and never both wild-type or both mutant) in the data from the sterile sector demonstrating that these alleles are encoded by the two homologous chromosomes. Taken together, this further demonstrates that the Q150* mutation in *Sobic.004G270900* is the molecular basis of the *ms8-1* mutant phenotype and the novel male sterile allele *ms8-2* carries two linked mutations resulting in P98S and Q111* changes in *Sobic.004G270900*.

### Identification and engineering of additional *ms8* mutant alleles

We also used CRISPR/Cas9-based gene editing to make additional alleles of *Sobic.004G270900* in Tx430, a different genetic background. Tx430 is a sorghum line that is more amenable to transformation (Howe *et al.* 2006). We used a gene editing toolkit

(Čermák *et al.* 2017) to express four gRNAs designed to target *SbiRTX430.04G283300*, the Tx430 homolog of *Sobic.004G270900*. We obtained two independent gene edited events, designated *ms8-3* and *ms8-4*, in the $T_1$ generation after removal of Cas9 (Fig. 6). *ms8-3* harbored a 205 bp deletion of coding sequence spanning gRNA2 to gRNA4 within exon 2 of *SbiRTX430.04G283300*. Similarly, *ms8-4* had a 416 bp deletion spanning from gRNA3 to gRNA4 in the same exon. Notably, both *ms8-3* and *ms8-4* exhibited a clear male sterile phenotype, phenocopying the *ms8-1* allele in BTx623.

We also searched a sequence-indexed sorghum mutant population generated previously (Addo-Quaye *et al.* 2018; Simons *et al.* 2022) for additional alleles at *MS8*. One accession (PI 678336) carried a nonsense mutation (Q123*) encoded by a C-to-T mutation at 61493928 bp in *MS8* (Fig. 6). We obtained 10 seeds of this accession from the USDA National Plant Germplasm System and planted them at the Purdue Agronomy Farm in West Lafayette in Summer 2024. This material was segregating for male sterility two out of eight heads from the progeny exhibited white anthers and failed to produce pollen. This allele is designated as *ms8-5*.

## Discussion

Seed EMS mutagenesis has been widely used to generate mutant alleles for functional genomics studies. A single mutant allele is often sufficient to strongly implicate a candidate gene but isolation and molecular identification of independent alleles that fail to complement is required to unequivocally assign a gene to a phenotype. As many genetic resources often return a single allele, publication of gene identification and function from mutant studies can be laborious and delayed. In this study, we demonstrate the utility of pairing sequencing with classic chemical mutagenesis to successfully identify and validate novel alleles in sorghum. We identified

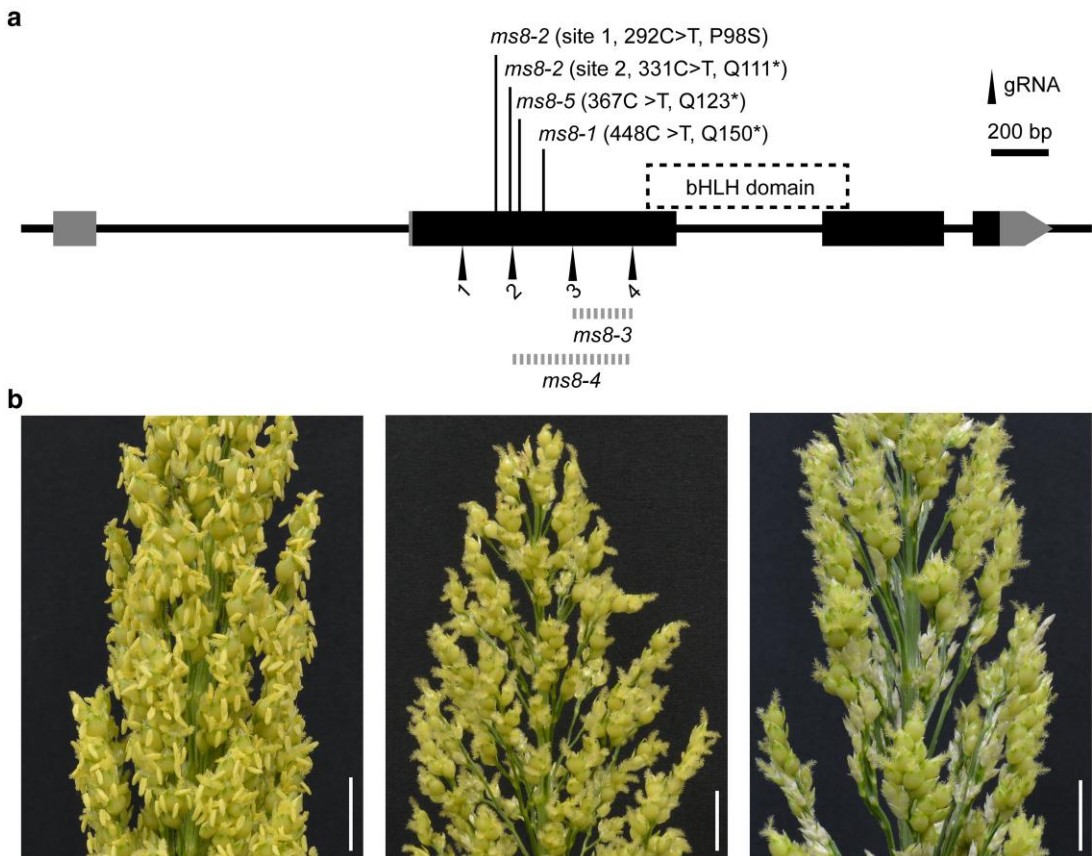

**Fig. 6.** *MS8* gene structures and *ms8* alleles. a) Schematic structure of the *MS8* gene. Dark rectangles represent protein-coding sequences, and gray rectangles represent the 5′ untranslated region (UTR) or 3′ UTR. The target sites of four gRNAs designed for *MS8* gene editing are indicated. A C-to-T mutation in the *MS8* locus leads to a stop codon gain in the *ms8-1* mutant. Two independent CRISPR-generated *ms8 alleles*, *ms8-3*, and *ms8-4*, harbor genomic deletions ranging from gRNA3 to gRNA4 and gRNA2 to gRNA4, respectively. Dashed rectangle represents the regions encoding bHLH protein domain. b) Male sterility phenotypes of *ms8* CRISPR lines *ms8-3* and *ms8-4*. Left: Tx430 wildtype control; middle: *ms8-3*; right: *ms8-4*. Scale bars represent 2 cm.

a conserved bHLH transcription factor as being necessary for male fertility and the locus responsible for the sorghum *ms8* phenotype. Our results highlight the application of targeted mutagenesis for accelerating gene discovery and validation, particularly when transformation resources are underdeveloped.

## Application of targeted seed EMS mutagenesis in genetic studies and sorghum breeding

Sorghum is an important agronomic crop that can withstand high temperatures and thrive on limited water (Prasad *et al.* 2021). Given the adaptability of sorghum across extreme climates, previous work has developed resources to study sorghum's resilience (Boyles *et al.* 2019; Xin *et al.* 2021). Many researchers lack access to transformation facilities and technology and regulatory hurdles for transgenic plants further limit use of gene editing to select countries. Sorghum transformation also requires time, specialized resources and expertise and is genotype-dependent (Casas *et al.* 1993; Sharma *et al.* 2020). As a result, an economical and effective method to generate novel alleles and validate candidate genes could accelerate the work of geneticists and breeders.

In our targeted mutagenesis screen, we used wildtype BTx623 as the pollen-parent. Therefore, mutagenesis of the heterozygous *MS8/ms8-1* material generated a mutant allele that was strictly derived from a wildtype BTx623 *MS8* allele. This targeted mutagenesis approach could have used any genetic background and allows researchers to target the genetic backgrounds of their choice for isolating novel alleles. This makes it possible to generate alleles in traditional varieties or advanced breeding stocks if these are intended targets. The ability to generate novel alleles directly in haplotypes of choice can streamline the breeding process by eliminating the need to transfer mutations from different genetic backgrounds. While background mutations introduced during targeted seed mutagenesis require subsequent backcrossing, this approach reduces linkage drag and allows breeders to focus on introgression of desired alleles into elite lines with fewer generations. Transformation procedures relying on somatic embryogenesis have eliminated the need for amenable genetic backgrounds for transformation, however, such procedures require specialized laboratory spaces, skills, regulatory approvals, and intellectual property licensing that can limit their utility in research and breeding. The targeted mutagenesis approach is a classical genetic approach that can be readily deployed by researchers across the globe.

In this study, we used this approach to identify a novel allele and validate a gene underlying male sterility, but our previous work in maize has used a similar experimental design to identify intragenic somatic suppressors of dominant alleles of a disease resistance locus (Karre *et al.* 2021). These experimental designs can be used for both recessive and dominant Mendelian traits where a phenotype can be readily observed on any visible plant (Candela and Hake 2008). Some important agronomic Mendelian traits that can be targeted using our approach include leaf color/

chlorophyll content, leaf waxiness, lignin composition (brown midrib), panicle branching, and seed color.

## Conservation of male sterility regulatory network across species

The *ms8-1* allele, caused by nonsense mutation in the sorghum ortholog of *bhlh122* from maize, exhibits defects in tapetum development (Xin *et al.* 2017). The tapetum, a specialized cell layer that surrounds the developing microspores (pollen mother cells) within the anther locules, provides nutrients and structural components necessary for pollen development and maturation (Wei and Ma 2023). This helps to regulate the timing and progression of pollen development, ensuring the proper formation of functional pollen grains. Male sterile mutants caused by tapetum defects have been reported in many plant species, including the *ms23, ms32, bhlh122-1*, and *bhlh51-1* mutants in maize (Nan *et al.* 2022). The genes *ms23, ms32, bhlh122*, and *bhlh51* encode four bHLH transcription factors that interact with each other to regulate tapetum and pollen development. Orthologs of these genes also regulate tapetum and pollen development in multiple angiosperm species including monocots and dicots (Niu *et al.* 2013; Ji *et al.* 2013; Zhu *et al.* 2015). The conservation of this bHLH-mediated regulatory mechanism across angiosperm lineages is remarkable but further research is needed to fully understand the extent and nature of this conservation. The integration of comparative genomics data alongside functional studies across greater diversity of plants is necessary to determine if this pathway is flower specific or participates in male fertility across the Angiosperms.

## Implications of male sterility for sorghum breeding and beyond

Male sterility is a trait that is valuable for hybrid seed production, as it prevents self-pollination and ensures cross-pollination. In sorghum, male sterility is a critical component of hybrid seed production and thus a major driver of grain yield. Although cytoplasmic male sterility is generally used in sorghum breeding programs, the molecular characterization of nuclear male sterility genes like *MS8* offers opportunities to develop innovative and flexible breeding systems including the engineering of advanced conditional male sterility mechanisms (e.g. chemical or environmentally inducible) (Zhou *et al.* 2014; Wang *et al.* 2021). Furthermore, the targeted seed EMS mutagenesis approach presented here could be used to recover novel alleles in any nuclear male sterility genes in sorghum within improved genetic backgrounds.

The recovery of novel alleles as mitotically cohesive sectors and coupling of screening and sequencing suggest that this approach could be valuable beyond species typically used for fundamental genetic studies. For example, chemical mutagenesis of clonally propagated species should similarly result in sector recovery when a novel allele fails to complement a mutant carried by a heterozygous individual. This opens the possibility of using EMS mutagenesis and DNA sequencing to provide multiple alleles in mutants in long-lived species, and species in which crossing is not facile. Working with mitotic sectors and recovering multiple independent alleles in sectors has the potential to expand the number of plant species amenable for gene function discovery. As such, using sectors to molecularly identify alleles can accelerate functional genetics studies in species with long generation times. For example, in poplar trees, cuttings from a heterozygous parent plant or heterozygous siblings of a recessive mutant can be mutagenized and affected sectors sequenced to identify genes with a novel heterozygous mutant, just as was done here for *ms8-2*. This would accelerate and potentiate the discovery of gene function by avoiding the necessity of multiple

generations beyond the generation used to recover the mutant to identify multiple alleles. Broadening the tools available to include sector approaches is particularly valuable for species with limited resources and therefore single alleles or those systems that are recalcitrant to genetic transformation.

## Data availability

The raw sequencing data from the BSA are available at NCBI under the BioProject PRJNA1134249. The raw paired-end sequencing reads and the BAM files from whole-genome sequencing of the pollen-fertile and pollen-sterile chimeric sectors containing *ms8-2* and *ms8-1* are available at NCBI under the BioProject (PRJNA1163370). Seeds of the germplasm described in this manuscript are available upon request to the corresponding authors.

Supplemental material available at GENETICS online.

## Acknowledgments

We thank Dr. Junpeng Zhang from Dr. Blake Meyers's lab at DDPSC for his suggestion in making phylogenetic trees of bHLH transcription factors. We would like to acknowledge Terry Beeler and his team at the Danforth Center Field Research Site and Rachel Stevens and the staff at the Agronomy Center for Research and Education, and Indiana Corn and Soybean Center at Purdue University for help with field experiments, especially field preparation and planting.

## Funding

This work was supported by funding from the US Department of Energy Biological and Environmental Research awards #DE-SC0020401 and #DE-SC0023305 to B.P.D. and A.L.E.; R.S.K. was supported by a Postdoctoral Fellowship from the USDA National Institute of Food and Agriculture award #2022-67012-36601.

## Conflicts of interest

The author(s) declare no conflicts of interest.

## Author contributions

Y.X., R.S.K., B.P.D., and A.L.E. designed the research, Y.X., R.S.K., and Z.W. performed experiments, Y.X., R.S.K., B.P.D. analyzed data, Y.X., R.S.K., B.P.D., and A.L.E. wrote the manuscript. All authors read and approved the manuscript.

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

Editor: T. Juenger