## [Peer Review File · Genetics]

Targeted seed EMS mutagenesis reveals a bHLH transcription factor underlying male sterility in sorghum

Yuguo Xiao, Rajdeep Khangura, Zhonghui Wang, Brian Dilkes, and Andrea Eveland

NOTE: The reviews and decision letters are unedited and appear as submitted by the reviewers.

In extremely rare instances and as determined by a Senior Editor or the EIC, portions of a review may be redacted. If a review is signed, the reviewer has agreed to no longer remain anonymous.

The review history appears in chronological order.

Review Timeline:

Submission Date:	2024-10-06
Editorial Decision:	2024-11-13
Revision Received:	2025-01-12
Accepted:	2025-01-14

November 13, 2024

RE: GENETICS-2024-307506

Dear Dr. Eveland:

I am pleased to accept your manuscript entitled "Targeted seed EMS mutagenesis reveals a bHLH transcription factor underlying male sterility in sorghum" for publication in GENETICS, pending minor revision.

Please submit your revision along with a response to the reviewers' concerns and suggestions, which can be viewed at the bottom of this email. In general, the reviewers found the analyses and results compelling in the context of trait discovery. However, there was some discussion about the utility of the approach for crop improvement and breeding. It would be helpful to clarify strategies that may be useful, especially in the context of comments by Reviewer 1. Reviewer 2 notes a number of points of the manuscript where additional information and clarification would be helpful. I expect this can be done within 30 days.

Upon resubmission, please include:

1. A clean version of your manuscript;
2. A marked version of your manuscript in which you highlight significant revisions carried out in response to the major points raised by the editor/reviewers (track changes is acceptable if preferred);
3. A detailed response to the editor's/reviewers' comments and to the concerns listed above. Please reference line numbers in this response to aid the editors.

Additionally, please ensure that your revision is formatted for GENETICS: <https://academic.oup.com/genetics/pages/general-instructions>.

Follow this link to submit the revised manuscript: Link Not Available

Thank you for submitting your research to Genetics.

Sincerely,
Tom Juenger
Series Editor, Plant Genetics & Genomics

Approved by:
Kate O'Connor-Giles
Senior Editor
GENETICS

Reviewer comments:

Reviewer #1 :

Knowledge of the genetics underlying traits is important for developing improved crops using modern breeding practices. Chemical mutagenesis and CRISPR-Cas biotechnology tools can generate DNA changes to study gene function. In this study, Xiao et al. utilized BSA-sequencing to identify sorghum bHLH transcription factor on chromosome 4 as a candidate gene controlling sorghum male sterility in the previously reported ms8 mutant. To test the role of this gene in male sterility, the authors conducted targeted seed EMS mutagenesis of heterozygous seed (Ms8ms8). The progeny of the heterozygous M1 seed should be mostly fertile as the plant contains a functional Ms8 allele. The authors predicted the M1 progeny would contain a partially sterile head in the case of a mutation in the WT Ms8 allele (Ms8*ms8). Of the 2500 plants, only one plant with partially sterile panicles was observed. WGS of the fertile and sterile sectors of the panicle showed an EMS-induced mutation in the sterile sector, validating the role of the bHLH transcription factor in sorghum pollen development. Additionally, the authors conducted CRISPR-Cas9 gene editing to demonstrate that ms8 null alleles were responsible for sorghum male sterility. Overall, the experimental design and data analyses clearly demonstrated the role of ms8 in pollen development.

Minor comments:

In lines 419-421, the authors suggested an application of targeted seed mutagenesis: 'The ability to generate novel alleles in haplotypes of choice can speed up breeding and trait introgression by reducing the number of backcrossing.' As targeted seed

mutagenesis also generates several mutations in the background, purging the background mutations by backcrossing will slow down the breeding process. I believe targeted seed mutagenesis cannot significantly speed up the breeding process.

Lines 431-433, 'Some of the important agronomic Mendelian traits that can be easily targeted using our approach include leaf, shoot and inflorescence architecture, lignin composition (brown midrib), and disease resistance.' It is unclear what inflorescence architecture traits can be easily identified using the sectors.

Lines 455-457: 'In sorghum, male sterility is a key component for hybrid seed production, a major driver of increased grain yield. Our demonstration of the molecular identity of the MS8 gene offers potential targets for manipulation in breeding programs.' Sorghum hybrid seed production uses cytoplasmic male sterility in producing hybrids. Nuclear male sterility is not used in hybrid seed production as the male sterility genes are recessive and cannot produce only sterile inbreds that can be used as females in hybrid seed production. It is unclear how the molecular identity of the MS8 gene offers potential targets for manipulation in breeding programs.

Reviewer #2 : attached

Authors report an in-built complementation approach and prove its applicability in plant functional genomics. The approach should have a broad impact for its potential use in various plant species, especially those lacking efficient genetic transformation. Using this approach, they demonstrated that *Sobic.004G270900*, which encodes one bHLH transcription factor, accounts for male fertility in sorghum, similar as found in rice, maize and Arabidopsis. I favor its publication with further revision.

Major comments:

1. Explain why not performing a dose test with EMS (Line 203), unless a dose test is not necessary for the reported approach.
2. Lines 269-277: “Among the 48 and 101 homozygous SNPs detected in *ms8-pool1* and *ms8-pool2*, 28 were shared in both pools.” Did you look at their genome distribution of the 48 and 101 SNPs? Whether the 28 shared SNPs are located in an overlapping region? If so, you actually can narrow down the *ms8-1* interval to a smaller region containing less genes than 968. Their chromosome distribution should be addressed.
3. Lines 296-297: “We pollinated male sterile *ms8-1* homozygotes with pollen from *MS8/ms8-1* heterozygotes.” Please explain why not using pollen from *MS8/MS8* homozygotes; what are the benefits of using the *MS8/ms8-1* heterozygotes for pollen donor?
4. Lines 318-320: “We identified a single fertile sorghum head with a sector displaying white anthers, resembling *ms8-1/ms8-1* homozygotes, on an otherwise fertile head (Figure 4a).” Apparently, that targeted mutation phenotype is very low, not in agreement with the statement of Lines 300-302. You should discuss what is the cause of this. I would guess that if you treat only the *MS8/ms8-1* genotype for EMS mutagenesis, you should be able to recover some fully sterile mutants. But in the current design, even there was a chance, there is no way to differentiate from the male-sterile *ms8-1/ms8-1* individuals which were segregated from the current cross.
5. Discussion: Avoiding reintroducing the results.

Minor comments:

1. 4000 M_1 seeds were planted to generate ~ 2500 M_1 plants. Were they planted in pots or directly in field? Anyhow, the surviving rate of M_1 s indicates that EMS dose was kind of low for an effective mutagenesis.
2. Figure 2: Phylogeny with full-length or trimmed bHLH proteins?

Line 35: “the sorghum ortholog of”

Lines 58-60: Reference does not justify the statement, or confusing statement.

Line 63: don't understand “but are often limited in size”

Line 80: “the F_1 generation”, here 1 in F_1 should be subscript, also all other similar expressions (e.g. lines 137, 139, and 244) in this manuscript.

Lines 137-139: confusing, hard to understand.

Line 142: “from male sterile and male fertile plants, respectively,”

Line 153: BTx623 (Sbicolor_454_v3.0.1, REFERENCE)

Response to Reviewers

We were delighted by the level of reviewer interest and appreciation of the work. We appreciate the comments and suggestions provided by the two reviewers as they assisted us in improving the overall quality and clarity of the manuscript. We revised several statements and paragraphs as described in the responses below for reader clarity. We also streamlined the Discussion and added Supplementary Table 4 based on suggestions from the reviewers.

Below is a point-by-point response to reviewer comments and suggestions. We submitted a revised manuscript file that incorporates changes based on these suggestions, as well as the original manuscript file marked up with track changes.

Please note, when using track changes to the original document the line numbers were altered significantly. In this response document, **line numbers refer to the revised, clean document. I tried to include both sets, but everything changed again when the pdf was created for the tracked changes version.

Response to Reviewer 1:

Knowledge of the genetics underlying traits is important for developing improved crops using modern breeding practices. Chemical mutagenesis and CRISPR-Cas biotechnology tools can generate DNA changes to study gene function. In this study, Xiao et al. utilized BSA-sequencing to identify sorghum bHLH transcription factor on chromosome 4 as a candidate gene controlling sorghum male sterility in the previously reported ms8 mutant. To test the role of this gene in male sterility, the authors conducted targeted seed EMS mutagenesis of heterozygous seed (Ms8ms8). The progeny of the heterozygous M1 seed should be mostly fertile as the plant contains a functional Ms8 allele. The authors predicted the M1 progeny would contain a partially sterile head in the case of a mutation in the WT Ms8 allele (Ms8*ms8). Of the 2500 plants, only one plant with partially sterile panicles was observed. WGS of the fertile and sterile sectors of the panicle showed an EMS-induced mutation in the sterile sector, validating the role of the bHLH transcription factor in sorghum pollen development. Additionally, the authors conducted CRISPR-Cas9 gene editing to demonstrate that ms8 null alleles were responsible for sorghum male sterility. Overall, the experimental design and data analyses clearly demonstrated the role of ms8 in pollen development.

Minor comments:

In lines 419-421, the authors suggested an application of targeted seed mutagenesis: 'The ability to generate novel alleles in haplotypes of choice can speed up breeding and trait introgression by reducing the number of backcrossing.' As targeted seed mutagenesis also generates several mutations in the background, purging the background mutations by backcrossing will slow down the breeding process. I believe targeted seed mutagenesis cannot significantly speed up the breeding process.

Response: Indeed, EMS mutagenesis introduces background mutations that must be purged by backcrossing. However, it introduces many orders of magnitude fewer mutations than a cross to a different genetic background. The ability to generate novel alleles on the chromosome from a background of interest by targeted mutagenesis would ultimately reduce linkage drag during trait integration. One can envision mutagenizing hybrids between a genotype of interest and a known mutant individual, and screening for desired mutant phenotypes. In this case, the new allele would be generated on the chromosome from the desired background. While backcrossing is necessary to remove unintended mutations, this is true for all forward genetic approaches.

We revised this in the manuscript to clarify this where lines 417-422 now read: *“The ability to generate novel alleles directly in haplotypes of choice can streamline the breeding process by eliminating the need to transfer mutations from different genetic backgrounds. While background*

mutations introduced during targeted seed mutagenesis require subsequent backcrossing, this approach reduces linkage drag and allows breeders to focus on introgression of desired alleles into elite lines with fewer generations."

Lines 431-433, 'Some of the important agronomic Mendelian traits that can be easily targeted using our approach include leaf, shoot and inflorescence architecture, lignin composition (brown midrib), and disease resistance.' It is unclear what inflorescence architecture traits can be easily identified using the sectors.

Response: We appreciate this comment since we can certainly be more specific here. We revised this now on lines 433-435 to read: *"Some important agronomic Mendelian traits that can be targeted using our approach include leaf color/chlorophyll content, leaf waxiness, lignin composition (brown midrib), panicle branching and seed color."*

Lines 455-457: 'In sorghum, male sterility is a key component for hybrid seed production, a major driver of increased grain yield. Our demonstration of the molecular identity of the MS8 gene offers potential targets for manipulation in breeding programs.' Sorghum hybrid seed production uses cytoplasmic male sterility in producing hybrids. Nuclear male sterility is not used in hybrid seed production as the male sterility genes are recessive and cannot produce only sterile inbreds that can be used as females in hybrid seed production. It is unclear how the molecular identity of the MS8 gene offers potential targets for manipulation in breeding programs.

Response: We appreciate this insightful comment. We recognize that cytoplasmic male sterility is predominantly used in sorghum hybrid seed production. However, exploring the molecular basis of nuclear male sterility genes like *MS8* can provide unique opportunities to develop innovative and flexible breeding systems, including the engineering of advanced conditional male sterility mechanisms (e.g., chemical or environmentally inducible systems).

We clarified this and lines 457-462 now read: *"In sorghum, male sterility is a critical component of hybrid seed production and thus a major driver of grain yield. Although cytoplasmic male sterility is generally used in sorghum breeding programs, the molecular characterization nuclear male sterility genes such as MS8 offers opportunities to develop innovative and flexible breeding systems including the engineering of advanced conditional male sterility mechanisms (e.g., chemical or environmentally inducible systems) (Zhou et al. 2014; Wang et al. 2021)."*

Response to Reviewer 2:

Authors report an in-built complementation approach and prove its applicability in plant functional genomics. The approach should have a broad impact for its potential use in various plant species, especially these lacking efficient genetic transformation. Using this approach, they demonstrated that *Sobic.004G270900*, which encodes one bHLH transcription factor, accounts for male fertility in sorghum, similar as found in rice, maize and Arabidopsis. I favor its publication with further revision.

Major comments:

1. Explain why not performing a dose test with EMS (Line 203), unless a dose test is not necessary for the reported approach.

Response: This was the method for this experiment. We've previously mutagenized Tx623 on multiple occasions and this concentration was chosen based on our prior experiences to balance mutation density and the induction of developmental abnormalities in the treated material. Indeed, we could have tried multiple doses in this experiment to spread some risk around but opted to maximize field space. We used

a concentration and treatment duration that has worked well for us in multiple years and worked for us here.

2. Lines 269-277: "Among the 48 and 101 homozygous SNPs detected in *ms8*-pool1 and *ms8*-pool2, 28 were shared in both pools." Did you look at their genome distribution of the 48 and 101 SNPs? Whether the 28 shared SNPs are located in an overlapping region? If so, you actually can narrow down the *ms8-1* interval to a smaller region containing less genes than 968. Their chromosome distribution should be addressed.

Response: We appreciate this suggestion. The genome distribution of the 48 and 101 SNPs from the two mutant pools is plotted in Figure 1c. We do want to note that the 968 genes within the initial *ms8-1* mapping interval (56–64 Mbp on Chr4) represent all annotated genes in this region, not just those containing EMS-induced SNPs. As described in the manuscript, only 5 variants alter protein coding. The 28 shared SNPs are indeed located in an overlapping region 58.3–62 Mbp, which reduces the candidate interval by 4 Mbp. This reduced interval contains a total of 445 genes.

We added a supplemental table (Supplemental Table 4) with detailed information about the 28 shared SNPs, including their chromosome coordinates, wild-type allele, and mutant allele. We also added a sentence in the manuscript (lines 279-280) explaining this: "*These 28 shared SNPs were located in an overlapping region that narrowed the candidate interval to 58.3-62 Mbp (Supplementary Table 4).*"

3. Lines 296-297: "We pollinated male sterile *ms8-1* homozygotes with pollen from *MS8/ms8-1* heterozygotes." Please explain why not using pollen from *MS8/MS8* homozygotes; what is the benefits of using the *MS8/ms8-1* heterozygotes for pollen donor?

Response: This is simply what we had available and not intentionally done for the sake of this experiment. This is how we propagate the nuclear male steriles and maintain seed stocks with higher frequency homozygotes. Thus, these were the seeds we had in abundance and were used for EMS mutagenesis.

4. Lines 318-320: "We identified a single fertile sorghum head with a sector displaying white anthers, resembling *ms8-1/ms8-1* homozygotes, on an otherwise fertile head (Figure 4a)." Apparently, that targeted mutation phenotype is very low, not in agreement with the statement of Lines 300-302. You should discuss what is the cause of this. I would guess that if you treat only the *MS8/ms8-1* genotype for EMS mutagenesis, you should be able to recover some fully sterile mutants. But in the current design, even there was a chance, there is no way to differentiate from the male-sterile *ms8-1/ms8-1* individuals which were segregated from the current cross.

Response: EMS mutagenesis of sorghum seeds generally creates chimeric heads. We typically recover, for dominant mutants, chimeric heads owing to the number of meristem cells in the seed that contribute to the head in sorghum. Segregation of recessive mutant phenotypes in M_2 generations are not 3:1, but rather 7:1 and lower, again consistent with chimeric heads. So, we do not think that we are missing a preponderance of *ms8* KO alleles that resulted from the full conversion of a heterozygous plant to a fully sterile head. It is true that if such a sector arose, in which *MS8* was knocked out and the cell encoding the mutation were to take over the entire SAM, we would miss it. Since we have over the years observed many hundreds of primary inflorescence heads with chimeric display of phenotypes and measured M_2 recessive phenotype segregation ratios from seed EMS treatments in sorghum we think this is unlikely. Our recovery of a single mutation at *ms8* in this experiment does not contradict the statement "*The resulting M_1 population of ~2,500 plants displayed varying degrees of spontaneous, likely dominant,*

*somatic mutated sectors on their leaves, confirming the efficacy of this seed EMS mutagenesis". There are many more cells in the SAM that contribute to leaves, but only a few that contribute to the head. There are also many genes that result in dominant foliar phenotypes. As a result, EMS mutagenesis affecting any of these genes in the much larger population of SAM cells transitioning into leaves results in a higher frequency of somatic mutations on the leaves than KO alleles in the head. In addition, we cannot rule out that we, in our care to not have too many false positives, missed smaller inflorescence sectors. We recovered one *ms8* mutation, but we indeed may have missed a few.*

5. Discussion: Avoiding reintroducing the results.

Response: We really appreciate this feedback and see now where this is quite repetitive. We revised the first paragraph of the Discussion to remedy this. Lines 393-400 now read: *"As many genetic resources often return a single allele, publication of gene identification and function from mutant studies can be laborious and delayed. In this study, we demonstrate the utility of pairing sequencing with classic chemical mutagenesis to successfully identify and validate novel alleles in sorghum. We identified a conserved bHLH transcription factor as being necessary for male fertility and the locus responsible for the sorghum *ms8* phenotype. Our results highlight the application of targeted mutagenesis for accelerating gene discovery and validation, particularly when transformation resources are underdeveloped."*

Minor comments:

1. 4000 M₁ seeds were planted to generate ~ 2500 M₁ plants. Were they planted in pots or directly in field? Anyhow, the surviving rate of M₁s indicates that EMS dose was kind of low for an effective mutagenesis.

Response: The 4,000 M₁ seeds were planted directly in the field and this is indicated in the Materials and Methods. EMS dose and treatment times were selected so as not to generate too much developmental abnormality that would impact our ability to phenotype heads by visual screening. The EMS dose that we used resulted in a loss-of-function *ms8* allele that we were able to identify by visual screening of heads.

2. Figure 2: Phylogeny with full-length or trimmed bHLH proteins?

Response: The phylogeny was constructed using trimmed bHLH protein sequences, focused on conserved regions within the bHLH domains to ensure our analysis was based on homologous regions. We clarified this in the Materials and Methods (lines 195-197): *"Protein alignments were trimmed using trimAL v1.4.1 (Capella-Gutiérrez et al. 2009) with parameters -gt 0.9 -cons 60 -w 3 to remove non-homologous regions outside the bHLH domains."*

Line 35: "the sorghum ortholog of"

Response: This was changed as suggested. Thank you.

Lines 58-60: Reference does not justify the statement, or confusing statement.

Response: We appreciate this comment to make sure the introduction to the manuscript is not confusing. We revised the sentence, which now should make sense in relation to the reference and added two more references. Lines 58-60 now read: *"Mutant populations are valuable resources for functional genomics, enabling causative mutations to be linked to phenotypes of interest through mapping and/or sequencing (Jiang and Ramachandran 2010, Gupta et al., 2023, Xiong et al., 2023)."*

Line 63: don't understand "but are often limited in size"

Response: We revised this sentence on lines 61-64 to read: "*Reverse genetics resources can be used to validate gene candidates identified through forward genetics or other methods but the number of lines sequenced is often limited and multiple loss-of-function alleles may not be recovered for a given gene.*"

Line 80: "the F₁ generation", here 1 in F₁ should be subscript, also all other similar expressions (e.g. lines 137, 139, and 244) in this manuscript.

Response: We appreciate this observation. Subscripts have incorporated for relevant expressions (e.g., F₁, T₁) throughout the manuscript.

Lines 137-139: confusing, hard to understand.

Response: We mapped *ms8-1* two times independently while using it as the female parent in mapping unrelated other mutant alleles. Thank you for calling attention to this. We agree that the sentence is unclear and revised it for clarity. Lines 137-139 now read: "*Two independent ms8-1 F₂ mapping populations were generated by crossing pollen from other EMS mutagenized BTx623 lines (Addo-Quaye et al. 2018) onto ms8-1 heads, and selfing the F₁ MS8/ms8-1 plants.*"

Line 142: "from male sterile and male fertile plants, respectively,"

Response: This has been fixed as suggested.

Line 153: BTx623 (Sbicolor_454_v3.0.1, REFERENCE)

Response: We included the cited reference here at first mention of the reference genome at line 153.

January 14, 2025

RE: GENETICS-2024-307506R1

Dr. Andrea L. Eveland
Donald Danforth Plant Science Center
Plant Developmental Genetics
975 N. WARSON ROAD
ST. LOUIS, Missouri 63108

Dear Dr. Eveland:

Congratulations, your manuscript entitled "Targeted seed EMS mutagenesis reveals a bHLH transcription factor underlying male sterility in sorghum" is accepted for publication in GENETICS! Many thanks for submitting your research to the journal.

To Proceed to Publication:

1. Format your article according to GENETICS style: <https://academic.oup.com/genetics/pages/general-instructions>

2. Ensure that you comply with data and community resource citation guidelines:

<https://academic.oup.com/genetics/pages/general-instructions#Data-Policy>

3. Upload your final files at <https://genetics.msubmit.net>

4. Add oupsupport@scipris.com and genetics.oup@novatechset.com (or the domains @scipris.com and @novatechset.com) to your email program's "safe senders" list. You will be contacted by both at various points during the production process.

Notes:

- Your currently-accepted manuscript (unedited, as submitted, reviewed, and accepted) will be published at GENETICS and deposited into PubMed as an Advance Access article. Notify sourcefiles@thegsajournals.org before signing your license if you do not wish to publish your article via Advance Access.

- We invite you to submit an original color figure related to your paper for consideration as cover art. Please email your submission to the editorial office or upload it with your final files. You can submit a small-sized image for evaluation, and if selected, the final image must be a TIFF file 2513px wide by 3263px high (8.375 by 10.875 inches; resolution of 600ppi). Please avoid graphs and small type.

- After files are sent to Oxford University Press we use SciPris to manage article licensing and payment. If you do not have a SciPris account, you will receive an email from no-reply@scipris.com to sign up to use Oxford University Press' author portal. After logging in, follow the online instructions to sign your license and arrange any payment due.

If you have any questions or encounter any problems while uploading your accepted manuscript files, please email the editorial office at sourcefiles@thegsajournals.org.

Sincerely,
Tom Juenger
Series Editor, Plant Genetics & Genomics

Approved by:
Kate O'Connor-Giles
Senior Editor
GENETICS